# Portuguese Nurses’ Stress, Anxiety, and Depression Reduction Strategies during the COVID-19 Outbreak

**DOI:** 10.3390/ijerph18073490

**Published:** 2021-03-27

**Authors:** Lara Guedes de Pinho, Francisco Sampaio, Carlos Sequeira, Laetitia Teixeira, César Fonseca, Manuel José Lopes

**Affiliations:** 1S. João de Deus School of Nursing, University of Évora, 7000-811 Évora, Portugal; cfonseca@uevora.pt (C.F.); mjl@uevora.pt (M.J.L.); 2Comprehensive Health Research Centre (CHRC), 7000-811 Évora, Portugal; 3Faculty of Health Sciences, University Fernando Pessoa, 4200-253 Porto, Portugal; fsampaio@ufp.edu.pt; 4“NursID: Innovation & Development in Nursing”, Center for Health Technology and Services Research, 4200-450 Porto, Portugal; carlossequeira@esenf.pt; 5Nursing School of Porto, 4200-072 Porto, Portugal; 6Abel Salazar Institute of Biomedical Sciences, University of Porto, 4050-313 Porto, Portugal; laetitiateixeir@gmail.com; 7“AgeingC: AgeingCluster”, Center for Health Technology and Services Research, 4200-450 Porto, Portugal

**Keywords:** anxiety, COVID-19, depression, mental health, mental health nurses, nurses, Portugal, stress

## Abstract

The COVID-19 pandemic has contributed to mental health problems worldwide. Nurses are particularly prone to stress because they directly care for individuals with suspected or confirmed cases of COVID-19. The aims of this study were (a) to explore the association between the mental health promotion strategies used by nurses during the COVID-19 outbreak and their symptoms of depression, anxiety, and stress; (b) to compare the symptoms of depression, anxiety, and stress of mental health nurses to those of non-mental health nurses; and (c) to compare the frequency of use of mental health strategies of mental health nurses to those of non-mental health nurses. A cross-sectional study was conducted with a sample of 821 nurses. Univariate and multivariate regression models were developed to identify potential protective factors of depression, anxiety, and stress. The chi-square test was also used to compare the use of strategies among mental health and non-mental health nurses. Portuguese nurses demonstrated high symptoms of depressive symptoms, stress, and anxiety. Healthy eating, physical activity, rest between shifts, maintaining social contacts, verbalizing feelings/emotions, and spending less time searching for information about COVID-19 were associated with better mental health. Mental health nurses had less depression, anxiety, and stress, and used more strategies to promote mental health than other nurses. We consider it important to promote nurses’ mental health literacy by encouraging them to develop skills and strategies aimed at improving their resilience and ability to deal with difficult situations while caring for the population.

## 1. Introduction

Healthcare professionals are more exposed to traumatic events such as the suffering and death of patients due to direct contact with individuals infected by COVID-19 [1], which can aggravate symptoms of anxiety [2]. The increasing number of individuals with confirmed or suspected cases, deficiency of protective equipment, work overload, media reporting, and the nonexistence of specific medications for the virus are risk factors for mental health problems in these professionals [3]. For all these reasons, health professionals are at a higher risk of mental health issues, and symptoms of stress, anxiety, and depression may develop [3].

Previous studies related to the SARS outbreak in 2003 showed that health professionals had higher symptoms of depression, anxiety, and stress one year after the outbreak, compared to other professionals [4], indicating that the repercussions on mental health were prolonged over time. Given the importance of protecting the mental well-being and health of these professionals over the long term, concerns about the mental health and psychological recovery of health professionals, who are on the frontlines caring for individuals with COVID-19, have started to emerge [3,5].

In the context of the COVID-19 pandemic, health professionals showed higher levels of anxiety, stress and depression [6,7] than those of the general population [3,8,9,10]. However, other studies have shown that the prevalence of anxiety and depression among health professionals and the general population in COVID-19 context was similar [11,12].

The caregiving process requires a capacity for discernment and critical reasoning, both interrelated [13] and disturbed in situations of stress and anxiety exacerbation. Nurses have an important role in providing healthcare during the pandemic and represent the largest number of health professionals in this sector [14]. Therefore, it is important and urgent to preserve their mental health.

It should be noted that although a recent meta-analysis indicated there were no differences in the prevalence of anxiety and depression among health professionals and the general population, it also concluded that nurses were at higher risk of developing anxiety and depression [11]. A study conducted with nurses showed that those who presented the greatest perceived threat of the pandemic had more anxiety, insomnia, depression, and social dysfunction [15]. This symptomatology was more frequent in nurses and health professionals who directly engaged in the diagnosis, treatment, or provision of nursing care to individuals with suspected or confirmed COVID-19 [3]. Other studies have reported higher symptoms of anxiety and higher prevalence in nurses, when compared with other health professionals [11,16]. In a preliminary Portuguese study, with a sample of 767 nurses, it was found that by April 7, 12.3% had been mobilized to another service, and 3.5% to another unit [17]. The average weekly working hours of the sample was 42 h, with higher symptoms of anxiety, stress, and depression in those professionals working longer hours. Another conclusion of this study was that the perception of low quantity and quality of personal protective equipment is associated with higher levels of depression, anxiety, and stress. Research has shown that unexpected changes in professional and family dynamics can have repercussions on mental health, causing symptoms of depression, anxiety, or stress [17]. Therefore, it is fundamental to address the mental health of nurses in the context of the COVID-19 pandemic.

Epidemiological information on the mental health of nurses and the factors that influence it during the COVID-19 pandemic are scarce, yet fundamental for the management of their actions in this context [18]. Given that the studies are not unanimous, and that the meta-analysis concludes that nurses are the group most at risk, then it seems important to strengthen the study of the mental health of these professionals so that in the future it will be possible to draw more robust conclusions regarding the impact that the pandemic had on their mental health. In addition, it is important to understand whether the levels of stress, anxiety, and depression are associated with the frequency of use of mental health promotion strategies.

The WHO published a set of strategies to promote and protect the mental health of health professionals, including taking breaks and resting between work shifts, eating a healthy diet, participating in physical exercise, avoiding the use of substances (tobacco, alcohol, or other drugs) to deal with stress, and being in contact with family and friends, albeit virtually [19].

Considering the aforementioned concerns, the aims of this study were (a) to explore the association between the mental health promotion strategies used by nurses during the COVID-19 outbreak and their symptoms of depression, anxiety, and stress; (b) to compare the symptoms of depression, anxiety, and stress of mental health nurses to those of non-mental health nurses; and (c) to compare the frequency of use of mental health strategies of mental health nurses to those of non-mental health nurses. For this purpose, the following hypotheses were proposed:(1)The use of strategies to promote mental health by nurses, in the context of the COVID-19 pandemic, is significantly associated with symptoms of depression, anxiety, and stress.(2)Being a mental health nurse is significantly associated with fewer symptoms of depression, anxiety, and stress, compared to a non-mental health nurse.(3)Being a mental health nurse is significantly associated with the use of more mental health promotion strategies, compared to a non-mental health nurse.

Hypothesis (1) is grounded on prior systematic reviews [20,21] that have shown that some mental health promotion strategies such as practicing physical exercise or engaging in relaxing activities present positive effects in the adult general population. On the other hand, hypotheses (2) and (3) are grounded on the specific competencies of Portuguese mental health nurses who, according to the Regulation No. 515/2018, demonstrate high self-awareness, as well as knowledge and ability to perform interventions that help promote mental health and prevent mental illness. Indeed, in Portugal, nurses finish their graduation and, after that, if they wish, they can specialize in a specific area. One of these areas is psychiatric and mental health nursing. Thus, mental health nurses develop specific competences, such as self-awareness and the ability to perform, for instance, psychoeducational, psychotherapeutic, and psychosocial interventions [22].

Finally, given that this study was conducted in Portugal, it seems relevant to discuss the geographical context of this country and to present data on the beginning and evolution of the pandemic. In 2019, Portugal had approximately 10.3 million residents [23]. The first case of COVID-19 infection appeared on 3 March 2020. The epidemiological evolution of the pandemic in the data collection period of this study was as follows: until 30 March 2020, at 12 a.m., Portugal had registered 7443 COVID-19 cases, 43 recovered cases, and 160 deaths; by 3 February 2021, there had been 726,321 confirmed COVID-19 cases, 534,384 recovered cases, and 12,757 deaths [24].

## 2. Materials and Methods

### 2.1. Design

This is a cross-sectional study that followed the Strengthening the Reporting of Observational Studies in Epidemiology guidelines. It was administered as an online questionnaire to nurses working in clinical practices in Portugal, to evaluate depression, anxiety, and stress, as well as strategies for promoting mental health. Nurses who were teleworking, those who did not perform functions in clinical practice, and those who were absent from work due to medical discharge or other reasons, were excluded from the study.

### 2.2. Data Collection

In total, 821 nurses working in clinical practice in Portugal were recruited for this study. We used the non-probabilistic snowball sampling method. The questionnaire was created using Google Forms and two members of the research team (FS and CS) sent the online questionnaire via email to all the nurses of their contact list who were working in healthcare settings. Nurses were asked to fill in the questionnaire and share it with other nurses in the same professional situation. Responses were obtained from nurses throughout the country. The exclusion criteria were nurses who were not working in clinical practice at the time of data collection.

A pretest of the online questionnaire was conducted with 10 nurses and no suggestions for change were reported. The questionnaire was considered understandable and easy to complete by all the nurses who were invited to participate in the pretest.

Data were collected from 31 March 2020 to 14 April 2020, a period in which Portugal was in a state of emergency, which was declared on 18 March 2020 [25], and in the mitigation phase of the COVID-19 pandemic.

The questionnaire consisted of four main sections, described in detail below: (1) sociodemographic data, (2) professional data, (3) data on the mental health promotion strategies used, and (4) measurement tools for depression, anxiety, and stress.

To characterize the sample, we collected sociodemographic data, such as gender, age, academic qualifications, marital status, and nationality, and professional data, such as where they worked and their specialty. Data related to the use of strategies to promote mental health were collected using a questionnaire, and data related to their mental health were assessed using the Depression Anxiety Stress Scale—short version (DASS-21).

This study was approved by two ethics committees. Data confidentiality was ensured by assigning a code to each participant, and no data was collected that could characterize the participant as to identify him/her. Participants could only access the questionnaire after giving their consent to participate.

### 2.3. Strategies of Mental Health Promotion

Mental health promotion strategies were assessed using a questionnaire developed by the research team for this study, based on the indications of the WHO for the promotion of mental health [19]. This questionnaire comprised nine questions, with each question corresponding to a strategy. The questionnaire assessed the frequency of use of the following strategies: break between work shifts; healthy eating; adequate water intake; physical activity; relaxing activities; recreational activities (e.g., reading, listening to music, watching movies/TV series); maintenance of social connections (while practicing social distancing); verbalization of feelings/emotions; and avoidance of information about COVID-19 from unreliable sources. The questions on the questionnaire were closed-ended and used an ordinal scale with the options never, rarely, sometimes, often, and always. The Cronbach’s alpha for this questionnaire was 0.77.

### 2.4. Depression Anxiety Stress Scale—Short Version (DASS-21)

The DASS-21 was developed to assess the symptoms of depression, anxiety, and stress. It was validated for the Portuguese population by Pais-Ribeiro, Honrado, and Leal (2004) [26]. The self-report DASS instrument comprises a set of three scales with seven items rated on a 4-point scale of severity/frequency that assesses the extent to which the individual experienced each state in the previous week [27]. The depression, anxiety, and stress scores are calculated by summing the scores of the respective items [27]. Each scale ranges from 0 to 21 points, and the higher the score, the more severe the symptoms of depression, anxiety, and/or stress [27]. Given that the DASS-21 reduced version has a score from 0 to 21, one can use these cut-off points by multiplying the scale value by two, which yields a score from 0 to 42 [27]. The Portuguese version of the DASS-21 has a Cronbach’s alpha of 0.85 for the depression subscale, 0.74 for the anxiety subscale, and 0.81 for the stress subscale. The tridimensional solution explains 50.35% of the variance (34.78% for the stress subscale, 9.14% for the depression subscale, and 6.47% for the anxiety subscale). There is a high correlation between the DASS-21 and DASS-42 (long version) subscales, with explained variances of 89%, 90%, and 96% for the stress, anxiety, and depression subscales, respectively [26].

### 2.5. Statistical Analysis

Descriptive statistics (mean, standard deviation, and absolute and relative frequency) were used according to the type of variable to characterize the study sample. Univariate and multivariate regression models were developed to identify potential protective factors (strategies used, time seeking information, mental health nursing specialty, and sociodemographic characteristics) of depression, anxiety, and stress. The chi-square test was also used to compare the use of strategies among groups (mental health and non-mental health nurses). The unadjusted models were performed as to identify the potential factors associated with each outcome, and to assist in the construction of the adjusted model. IBM SPSS Statistics version 24 (IBM Corp, Armonk, NY, USA) for Windows was used to perform the statistical analyses. The threshold for statistical significance was set at 0.05.

## 3. Results

### 3.1. Sociodemographic and Clinic Characteristics

The sample consisted of 821 participants (81.1% female) with an average age of 39.1 (SD = 9.6) years (range, 22–65 years), of whom 54.2% were nursing specialists and 18.5% were mental health nursing specialists. The dynamics of care delivery has changed, and nurses provide care for people suspected of or infected by COVID-19 in all health services. Therefore, the sample provided care for all patients, regardless of whether they were suspected or not of being infected with COVID-19. Moreover, 6.3% provided care in services created specifically to support people infected by COVID-19.

The respondents spent an average of 2.7 (2.2) hours per day seeking information about COVID-19.

The DASS-21 results indicated that, regarding symptoms within the normal range, 64.9% of participants presented symptoms of depression, 54.3% with anxiety, and 36.4% with stress. In relation to severe or extremely severe symptoms, 18.5% of participants presented anxiety, 10.9% stress, and 7.4% depression symptoms. The DASS-21 results are provided in Table 1.

### 3.2. Findings Related to the Hypotheses

Considering the unadjusted linear regression models, all factors tested were statistically significant.

The factors associated with fewer symptoms of depression were being older (*p* < 0.01); being a mental health nurse (*p* < 0.05); being male (*p* < 0.05); spending less time searching for information about COVID-19 (*p* < 0.01); resting often/always between work shifts, compared with only a few times (*p* < 0.01); eating healthily often/always, compared with never or rarely (*p* < 0.001) and sometimes (*p* < 0.01); frequently/always verbalizing feelings/emotions, compared with sometimes (*p* < 0.05); and often/always maintaining social connections, compared with sometimes (*p* < 0.05).

The factors associated with fewer symptoms of anxiety included being male (*p* < 0.001); being older (*p* < 0.001); being a mental health nurse (*p* < 0.05); spending less time searching for information about COVID-19 (*p* < 0.01); eating healthily frequent/always, compared to never or rarely (*p* < 0.05) and sometimes (*p* < 0.05); often/always resting between work shifts, compared to never/rarely (*p* < 0.05) and sometimes (*p* < 0.05); and often/always participating in physical activity, compared with never or rarely (*p* < 0.01).

The factors associated with fewer symptoms of stress included being male (*p* < 0.001); being older (*p* < 0.001); being a mental health nurse (*p* < 0.05); spending less time searching for information about COVID-19 (*p* < 0.001); eating healthily often/always, compared to never or rarely (*p* = 0.008) and sometimes (*p* = 0.001); and frequent/always resting between work shifts, compared with never/rarely (*p* < 0.05). The results of the adjusted models are shown in Table 2.

The results regarding the use of mental health promotion strategies by mental health nurses and non-mental health nurses are presented in Table 3. Mental health nurses used the following strategies more frequently: healthy eating (*p* < 0.01), adequate water intake (*p* < 0.05), relaxing activities (*p* < 0.001), recreational activities (*p* < 0.001), maintenance of social contacts (at a distance; *p* < 0.001), and verbalization of feelings/emotions (*p* < 0.01).

## 4. Discussion

The female sex showed higher symptoms of depression, anxiety, and stress. This result is corroborated by other studies in the general population [3,11,28,29,30]. In addition, an epidemiological study on the prevalence of depression between 1994 and 2014, which analyzed 90 studies from different countries (*n* = 1,112,573 adults), showed that the female sex is at higher risk of depression. Data from the present study are in line with these global data in periods outside the pandemic [31]. It is also reported in the literature that women have a higher prevalence of anxiety disorders [32]. Thus, the results of our study, with respect to sex, do not seem to be specific to nursing or the COVID-19 context, as they agree with studies conducted during non-pandemic periods and with the general population.

We now analyze the remaining results based on the hypotheses.

(1)
*The use of strategies to promote mental health by nurses, in the context of the COVID-19 pandemic, is significantly associated with symptoms of depression, anxiety, and stress.*


Fewer depressive symptomatologies were associated with the most frequent use of the following strategies: healthy eating, maintenance of social connections, rest between work shifts, and verbalization of feelings/emotions. Healthy eating and physical activity were related with lower anxiety. Stress was lower in those who more frequently ate a healthy diet and rested between work shifts. As we can observe, healthy eating was associated with fewer symptoms of depression, anxiety, and stress. Some studies indicate that the ingestion of vitamins and minerals is effective for reducing symptoms of depression, anxiety, and stress [33,34]. Therefore, considering that a healthy diet presupposes the consumption of vitamins and minerals, this association may be related to the higher consumption of these nutrients. In addition, other recent studies indicate that healthy nutrients (e.g., fructo-oligosaccharides, galacto-oligosaccharides, legumes, fruits, and nuts), help reduce symptoms of depression, anxiety, and stress in an individual [35]. Thus, the data are consistent with those in the literature. Regarding physical activity, another study conducted with nurses reported that physical inactivity is significantly positively related to anxiety symptoms [36], which is in line with the results of our study, given that nurses who participated in physical activity exhibited less anxiety symptoms. Resting between work shifts was associated with fewer symptoms of depression and stress. These data are in line with a scoping review that concluded that rest periods in the nursing profession positively influence mental well-being [37]. Maintenance of social connections was associated with fewer symptoms of depression, which is corroborated by studies that indicate that the absence of social contacts is heavily associated with current depression, and heightened vulnerability to future depression [38]. Another factor associated with fewer symptoms of depression was the verbalization of feelings or emotions, which is associated with improved well-being and decreased depressive symptoms, as observed in other studies [39].

Regarding relaxing and recreational activities and adequate water intake, we observed that these strategies did not have statistically significant differences in the regression model when we compared the frequency of their use by nurses and symptoms of depression, anxiety, and stress; this result was not expected and might be because there has been a pandemic-induced change in work routines and even in family dynamics. This may have led to the inability of nurses to relax as they usually would, even though some participated in relaxing and recreational activities. This might also be related to changes in how often these activities were performed during the pandemic, compared to previous period. As we do not have these data, we do not know whether the frequency decreased, was maintained, or increased. In any case, given the changes in work and family dynamics, it is likely that some nurses eventually decreased their frequency of participation in activities.

Our study showed that nurses who spent more time searching for information on the COVID-19 outbreak presented more symptoms of depression, anxiety, and stress than those who spent less time searching for such information. Similar data were found in another studies [8,11,30,40]. Therefore, our results are in line with the existing literature, and there seems to be a positive association between time spent watching news associated with the COVID-19 outbreak and symptoms of anxiety, stress, or depression.

Regarding the sources of information, this study did not obtain statistically significant results concerning the avoidance of access to information about COVID-19 from unreliable sources; therefore, further studies are needed to define whether the type of search source is associated with mental health. However, a recent meta-analysis concluded that having up-to-date and accurate health information seems to be a protective factor for psychological health [11].

(2)
*Being a mental health nurse is significantly associated with fewer symptoms of depression, anxiety, and stress, compared to a non-mental health nurse.*


Mental health nurses acquire a series of skills in their specialized training that allow them to have “high knowledge and self-awareness as individuals and nurses, through experiences and processes of self-knowledge and personal and professional development” [22]. It is expected that these competences provide them with the tools necessary to deal with adverse and unexpected situations, making them resilient, not only in the provision of mental health care to patients, but also in their mental self-care, implementing effective strategies for promoting mental health and preventing mental illness. A study conducted with nurses during the COVID-19 outbreak proved that resilience was associated with less anxiety related to COVID-19 [2].

In this study, nurses who were not specialized in mental health had more symptoms of depression, anxiety, and stress than those who were. This is a novel finding, with no major studies having obtained comparable results; we only found one study conducted in Hong Kong in 2015, which concluded that generalist nurses were 1.6 times more likely to suffer from anxiety symptoms and 2.2 times more likely to suffer from depression than mental health nurses [36].

(3)
*Being a mental health nurse is significantly associated with the use of more mental health promotion strategies, compared to a non-mental health nurse.*


Our results indicate that mental health nurses use the following strategies more frequently than nurses in other areas: healthy eating, adequate water intake, participation in relaxing and recreational activities, maintenance of social connections (while maintaining social distancing), and verbalization of feelings/emotions. However, since our results are not cause-effect, the fact that these professionals use mental health promotion strategies more often might also be because they have less symptoms of depression, anxiety, and stress.

A recent qualitative research study verified that support from family, friends, colleagues, organizations, and the public is crucial to promote nurses’ well-being during the COVID-19 pandemic. The same study suggests that institutions implement communication strategies between managers and nurses to promote the use of strategies to reduce stress and provide accurate information; provide support in basic daily needs; and promote psychosocial support, such as groups to express emotions [41]. Another recent study suggests psychoeducational interventions for nurses to promote coping strategies and reduce psychological symptomatology [15].

Notably, only 42.1% of our sample of mental health nurses worked in psychiatric services, while the remainder were distributed among non-psychiatric services. Some of these nurses were mobilized during the pandemic, which indicates that the differences observed may be related to the type of specific mental health training that the nurses underwent, which may have increased resilience to stressful situations, such as the COVID-19 pandemic, leading to improved resistance to depression, anxiety, and stress factors. As the literature indicates, cognitive competencies and problem-solving skills are important precursors of resilience [42], and despite these challenges, resilience allows nurses to deal with stressors in their work environment and maintain healthy and stable psychological functioning [43]. Thus, promoting resilience is crucial for nurses coping with work-related stress [44]. Furthermore, a strong support system is essential to promote resilience in nurses [41].

### Limitations

One of the limitations of the study is that it used the non-probabilistic snowball sampling method, which may have led to a potential sampling bias. However, despite this limitation, we were able to obtain a sample from different parts of the country. Furthermore, because the study involved self-reporting by participants, we should consider the risk of response bias. Another limitation is that the results come from a cross-sectional study, which makes it impossible to determine the temporal sequence of events. In other words, the results allow the identification of the association between variables but cannot determine cause-effect relationships. Another limitation is that, since the research was conducted online by using a snowball sampling, we cannot guarantee that all respondents were nurses. However, the questionnaire comprised a set of questions on professional data, so if the respondent were not a nurse, he/she would not be able to answer those questions. Finally, the fact that the set of measurements came from the same source (i.e., nurses) can lead to the possibility of a common variance bias.

## 5. Conclusions

Healthy eating, physical activity, rest between work shifts, maintenance of social connections, verbalization of feelings/emotions, and a shorter time spent searching for information about the COVID-19 pandemic were associated with better mental health in nurses. These strategies are accessible to everyone, because they are part of self-care and do not require substantial resources. Therefore, health professionals must promote the adoption of these strategies as self-care measures, for themselves and patients alike.

Therefore, it is extremely important to encourage nurses to adopt these mental health-promoting strategies, especially nurses with no specific training in mental health, as they tend to be more prone to depression, anxiety, and stress than their counterparts. This result leads us to believe that a greater focus on mental health literacy, directed at all nurses, is important to promote resilience and the consequent ability to solve problems which in turn may improve the quality of care provided to the population, especially in extreme situations such as a pandemic.

It is also important to understand whether symptoms of depression, anxiety, and stress decrease, increase, or remain the same in a post-pandemic context, to assess the possible need for intervention.

## Figures and Tables

**Table 1 ijerph-18-03490-t001:** Description of depression, anxiety, and stress (DASS-21).

	Mean (sd)	Range	%Normal	%Mild	%Moderate	%Severe	%Extremely Severe
DASS-21 Depression	4.00 (3.86)	0–21	64.9	14.9	12.8	4.5	2.9
DASS-21 Anxiety	4.18 (4.06)	0–21	54.3	9.6	17.5	7.7	10.8
DASS-21 Stress	7.32 (4.54)	0–21	36.4	36.9	15.7	8.0	2.9

sd: standard deviation.

**Table 2 ijerph-18-03490-t002:** Description of the survey demographics, professionals, mental health strategies, and mental health variables and the statistical results for the linear regression models.

	Adjusted Models								
	DASS-21 Depression			DASS-21 Anxiety			DASS-21 Stress		
	B (se)	95% CI	*p*	B (se)	95% CI	*p*	B (se)	95% CI	*p*
Sex: fem. [ref: male]	0.73 (0.32)	0.09–−1.37	0.025	1.35 (0.34)	0.68–2.03	<0.001	1.50 (0.38)	0.76–2.24	<0.001
Age	−0.04 (0.01)	−0.06–−0.01	0.005	−0.06 (0.01)	−0.09–−0.03	<0.001	−0.10 (0.02)	−0.13–−0.07	<0.001
Mental health specialty: yes [ref: no]	−0.76 (0.33)	−1.40–−0.12	0.020	−0.78 (0.34)	−1.45–−0.10	0.024	−0.74 (0.38)	−1.48–−0.01	0.048
Time searching for information	0.16 (0.06)	0.05–0.27	0.005	0.17 (0.06)	0.05–0.28	0.006	0.34 (0.07)	0.22–0.47	<0.001
Rest between work shifts [ref: often or always]									
Never or rarely	0.54 (0.42)	−0.30–1.37	0.206	1.34 (0.45)	0.46–2.23	0.003	0.98 (0.49)	0.02–1.94	0.047
Sometimes	0.79 (0.30)	0.20–1.38	0.009	0.69 (0.32)	0.07–1.32	0.030	0.68 (0.35)	−0.004–1.36	0.051
Eating healthy [ref: often or always]									
Never or rarely	1.71 (0.46)	0.80–2.62	<0.001	1.23 (0.49)	0.26–2.19	0.013	1.42 (0.53)	0.37–2.46	0.008
Sometimes	1.03 (0.32)	0.42–1.65	0.001	0.66 (0.33)	0.002–1.31	0.049	1.16 (0.36)	0.45–1.88	0.001
Adequate water intake [ref: often or always]									
Never or rarely	−0.08 (0.40)	−0.85–0.68	0.830	−0.02 (0.41)	−0.83–0.79	0.967	0.30 (0.45)	−0.58–1.18	0.503
Sometimes	0.19 (0.30)	−0.40–0.78	0.518	−0.22 (0.32)	−0.84–0.40	0.488	0.02 (0.35)	−0.66–0.69	0.964
Physical activity [ref: often or always]									
Never or rarely	0.68 (0.41)	−0.11–1.48	0.093	1.15 (0.43)	0.31–1.99	0.007	0.78 (0.47)	−0.14–1.70	0.097
Sometimes	0.21 (0.42)	−0.61–1.03	0.613	0.55 (0.44)	−0.32–1.41	0.215	0.33 (0.48)	−0.62–1.27	0.497
Relaxing activities [ref: often or always]									
Never or rarely	0.48 (0.45)	−0.40–1.36	0.281	0.32 (0.47)	−0.61–1.25	0.501	0.82 (0.52)	−0.19–1.83	0.113
Sometimes	0.08 (0.44)	−0.78–0.93	0.856	0.31 (0.46)	−0.59–1.22	0.500	0.72 (0.50)	−0.26–1.71	0.151
Recreational activities [ref: often or always]									
Never or rarely	0.55 (0.40)	−0.22–1.33	0.164	0.63 (0.42)	−0.19–1.46	0.133	0.79 (0.46)	−0.11–1.68	0.086
Sometimes	−0.19 (0.32)	−0.82–0.44	0.560	−0.20 (0.34)	−0.87–0.47	0.551	0.02 (0.37)	−0.71–0.75	0.966
Maintenance of social connections [ref: often or always]									
Never or rarely	1.23 (0.44)	0.37–2.08	0.005	0.81 (0.46)	−0.09–1.71	0.079	0.73 (0.50)	−0.25–1.72	0.144
Sometimes	0.14 (0.31)	−0.47–0.76	0.651	0.43 (0.33)	−0.22–1.08	0.199	0.28 (0.36)	−0.42–1.00	0.431
Verbalization of feelings/emotions [ref: often or always]									
Never or rarely	0.40 (0.35)	−0.30–1.09	0.263	−0.04 (0.37)	−0.77–0.70	0.918	0.09 (0.41)	−0.71–0.89	0.817
Sometimes	0.69 (0.33)	0.04–1.34	0.038	0.47 (0.35)	−0.22–1.15	0.183	0.52 (0.38)	−0.23–1.27	0.175
Avoidance of access to information from unreliable sources [ref: often or always]									
Never or rarely	0.16 (0.34)	−0.49–0.82	0.623	0.16 (0.36)	−0.54–0.86	0.659	−0.39 (0.39)	−1.15–0.37	0.313
Sometimes	0.33 (0.33)	−0.31–0.97	0.312	0.45 (0.34)	−0.22–1.13	0.190	−0.06 (0.38)	−0.79–0.68	0.880
	R^2^ = 19.4%; F = 8.643; *p* < 0.001			R^2^ = 18.5%; F = 8.143; *p* < 0.001			R^2^ = 23.2%; F = 10.852; *p* < 0.001		

B: regression coefficient. se: standard error. CI: confidence interval.

**Table 3 ijerph-18-03490-t003:** Strategies used for promoting mental health (*n* = 821).

	Never	Rarely	Sometimes	Often	Always	*p* Value *
	*n* (%)					
*Rest between work shifts*						
Mental health nurses	3 (2.0)	13 (8.6)	36 (23.7)	64 (42.1)	36 (23.7)	*p* = 0.196
Non-mental health nurses	27 (4.0)	52 (7.8)	190 (28.4)	260 (38.9)	140 (20.9)	
*Eating healthy*						
Mental health nurses	1 (0.7)	11 (7.2)	36 (23.7)	71 (46.7)	33 (21.7)	*p* = 0.006
Non-mental health nurses	4 (0.6)	77 (11.5)	203 (30.3)	283 (42.3)	102 (15.2)	
*Adequate water intake*						
Mental health nurses	2 (1.3)	20 (13.2)	51 (33.6)	55 (36.2)	24 (15.8)	*p* = 0.013
Non-mental health nurses	12 (1.8)	137 (20.5)	232 (34.7)	214 (32.0)	74 (11.1)	
*Physical activity*						
Mental health nurses	29 (19.1)	58 (38.2)	36 (23.7)	20 (13.2)	9 (5.9)	*p* = 0.253
Non-mental health nurses	167 (25)	223 (33.3)	176 (26.3)	75 (11.2)	28 (4.2)	
*Relaxing activities*						
Mental health nurses	13 (8.6)	52 (34.2)	54 (35.5)	26 (17.1)	7 (4.6)	*p* < 0.001
Non-mental health nurses	162 (24.2)	245 (36.6)	179 (26.8)	71 (10.6)	12 (1.8)	
*Recreational activities*						
Mental health nurses	4 (2.6)	20 (13.2)	51 (33.6)	56 (36.8)	21 (13.8)	*p* < 0.001
Non-mental health nurses	44 (6.6)	135 (20.2)	230 (34.4)	216 (32.3)	44 (6.6)	
*Maintenance of social connections (while social distancing)*						
Mental health nurses	1 (0.7)	14 (9.2)	49 (32.2)	62 (40.8)	26 (17.1)	*p* < 0.001
Non-mental health nurses	13 (1.9)	94 (14.1)	275 (41.1)	222 (33.2)	65 (9.7)	
*Verbalization of feelings/emotions*						
Mental health nurses	5 (3.3)	29 (19.1)	55 (36.2)	47 (30.9)	16 (10.5)	*p* = 0.003
Non-mental health nurses	43 (6.4)	184 (27.5)	222 (33.2)	180 (26.9)	40 (6.0)	
*Avoidance of access to information from unreliable sources about COVID-19*						
Mental health nurses	12 (7.9)	17 (11.2)	33 (21.7)	53 (34.9)	37 (24.3)	*p* = 0.661
Non-mental health nurses	47 (7.0)	86 (12.9)	138 (20.6)	211 (31.1)	187 (28)	

* Mann–Whitney U test. *n*: number of cases.

## Data Availability

Data available on request due to ethical restrictions.

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
