# Peer review of "Portuguese Nurses’ Stress, Anxiety, and Depression Reduction Strategies during the COVID-19 Outbreak"

_ijerph, 2021, doi:10.3390/ijerph18073490_

Round 1
Reviewer 1 Report
Thank you very much for the opportunity to review this manuscript. The manuscript documents Portuguese nurses demonstrated high symptoms of depressive symptoms, stress, and anxiety. The manuscript addressed a significant issue, and the article is well constructed. However, a plethora of studies has already examined this issue in different contexts. I have the following comments to improve this paper:
- At the end of the introduction, the authors highlight this paper's incremental contributions to the literature by looking at the highly cited studies e.g., https://scholar.google.com/scholar?hl=en&as_sdt=0%2C5&q=allintitle%3A+"mental+health"+and+covid-19"+and+nurses&btnG=.
- The authors cite the source on page 2 line 62-63.
- Authors provide details of the scales are developed by them or adopted from prior studies.
- Authors define the mental health nurses and how these nurses are different from non-mental health nurses?
- The authors also provide the detail of the pilot study results.
- I urge the authors to run a correlation analysis as it is a pre-requisite for the logistic regression analysis.
- Authors report R2 and F statistics of the regression models.
- In the discussion section authors provide the sub-headings for 4.1. Sociodemographic variables.
- Finally, the authors check the minor issues and typos.
Author Response
Dear reviewer,
Firstly, we would like to thank you very much for your analysis and for the opportunity to improve our paper. We found your recommendations extremely useful and we are sure they helped improve the overall quality of the manuscript.
We tried to address all your recommendations and to give response to all your comments.
- At the end of the introduction, the authors highlight this paper's incremental contributions to the literature by looking at the highly cited studies e.g., https://scholar.google.com/scholar?hl=en&as_sdt=0%2C5&q=allintitle%3A+"mental+health"+and+covid-19"+and+nurses&btnG=.
We added some information that was present in some of the highly cited studies on this topic and tried to make more clear the contribution of our study for the research on this field (in the “Introduction” section).
- The authors cite the source on page 2 line 62-63.
Thank you for your comment. We added the citation.
- Authors provide details of the scales are developed by them or adopted from prior studies.
We clarified that the questionnaire of mental health promotion strategies was developed by authors for this study; moreover, we clarified that the DASS-21 is an assessment tool which had previously been validated for the Portuguese population.
- Authors define the mental health nurses and how these nurses are different from non-mental health nurses?
We added this information to the “Introduction” section (“Indeed, in Portugal, nurses finish their graduation (…) psychotherapeutic and psychosocial interventions”).
- The authors also provide the detail of the pilot study results.
We added details of the pretest we carried out prior to the data collection, as well as of its results (in the “Data collection” section).
- I urge the authors to run a correlation analysis as it is a pre-requisite for the logistic regression analysis.
We did not carry out a logistic regression analysis in this study. However, we would like to thank you for your comment. We tried to clarify the statistical analysis we carried out in the “Statistical analysis” section.
- Authors report R2 and F statistics of the regression models.
We reported R2 and F statistics of the regression models in the “Results” section, table 2. (model 1: R2=19.4%; F=8.643; p<0.001; model 2: R2=18.5%; F=8.143; p<0.001; model 3: R2=23.2%; F=10.852; p<0.001).
- In the discussion section authors provide the sub-headings for 4.1. Sociodemographic variables.
We removed this sub-heading.
- Finally, the authors check the minor issues and typos.
We would like to thank you very much for your comments that helped us improving significantly our manuscript. We hope the changes that we made are in accordance with your suggestions.
A linguistic revision of the manuscript was made.
Reviewer 2 Report
Dear Authors, before publication, the article requires thorough improvement, details in the attachment.

Author Response
Dear reviewer,
Firstly, we would like to thank you very much for your analysis and for the opportunity to improve our paper. We found your recommendations extremely useful and we are sure they helped improve the overall quality of the manuscript.
We tried to address all your recommendations and to give response to all your comments.
Abstract: The aim of the study was very concise and difficult to understand.
We changed the aim of the study.
Line 18: “The aim of this study is…….”, or the aim of the study was?
We changed the verb tense.
There is no explanation about the statistical methods used in the study.
We added an explanation about the statistical methods to the Abstract.
Introduction: This is a story rather than a research paper. This part requires to be rewritten, taking into account the specifics of nurses' work at that particular time, including the workplace.
We tried to improve the “Introduction” by adding some information from previous studies, some of them focusing on the specifics of nurses’ work during the COVID-19 outbreak.
Purpose of the study - it is impossible to understand what the purpose of the study was.
We changed the aim of the study (in line with the change we made in the Abstract).
Methodology:
There is no description how volunteers were recruited for the study.
We add this in 2.2 section.
The research was conducted on-line, how can you be sure that only nurses had access to the questionnaire?
That is, indeed, a limitation of our study. We added that to the “Limitations” section (“Another limitation is that (…) able to answer those questions)”.
Please describe in detail the procedure for recruiting respondents, how the anonymity and confidentiality of the data have been ensured. What were the exclusion criteria from the study?
We added that information in the last paragraph of the section “Data collection”. We would like to reinforce the study was approved by two ethical committees.
How was the selection of the test sample made?
We used a snowball sampling, which does not allow the selection of the test sample. That limitation is addressed in the “Limitations” section.
Results: The results presented chaotically. There is no explanation of the abbreviations used below the tables.
We added an explanation of the abbreviations used below the tables. Moreover, we reorganize the “Results” section.
Discussion: Too long discussion, rather a description of the results than compared to other studies.
We reduced the “Discussion” section. Moreover, in order to make it more easy-reading, we organized that in accordance to the study hypotheses. In the current version of the manuscript, 20 studies are cited in the “Discussion” section to compare our findings with the ones of previous studies.
Conclusions are also too long and not very specific.
We reduced the “Conclusions” section.
The entire work is not written carefully, many typos and language errors, requires editorial correction, especially references and a list of references. The manuscript should be prepared in accordance with the guidelines for authors: https://www.mdpi.com/journal/ijerph/instructions.
We would like to thank you very much for your comments that helped us improving significantly our manuscript. We hope the changes that we made are in accordance with your suggestions.
A linguistic revision of the manuscript was made.
Reviewer 3 Report
This is, in my opinion, a high-quality study examining the association between the mental health promotion strategies used by nurses during the coronavirus disease (COVID-19) outbreak and their symptoms of depression, anxiety, and stress, and to compare the strategies and symptoms of mental health nurses and non-mental health nurses. This study has many strengths. First, the research is original and well defined. Second, the results provide an advance in current knowledge. Third, the article is well written and structured. Data are presented appropriately. Lastly, these results should be of interest for the readership of the Journal. Overall, I believe that this work merits being published. However, I have minor comments and concerns that I hope will be helpful.
1-) Congratulations for the questionnaire measuring the strategies of mental health promotion that was developed for this study. What was the Cronbach’s Alpha for this scale ? I suggest adding some information regarding its psychometric properties.
2-) A suggestion would be to add a theoretical background in order to support the hypothesis beyond the empirical support.
3-) The fact that the set of measurements came from the same source (i.e. nurses) leads to the possibility of a common variance bias could be added as a limitation. This is a suggestion.
I would like to congratulate the authors for the work !
Author Response
Dear reviewer,
Firstly, we would like to thank you very much for your analysis and for the opportunity to improve our paper. We found your recommendations extremely useful and we are sure they helped improve the overall quality of the manuscript.
We tried to address all your recommendations and to give response to all your comments.
1-) Thank you for your comment. We added the Cronbach’s alpha for the mental health promotion strategies to the “Strategies of mental health promotion” section.
2-) We added a theoretical background to support the hypotheses at the end of the “Introduction” section.
3-) Thank you for your suggestion. We added that, as a limitation, to the “Limitations” section.
Round 2
Reviewer 1 Report
Authors have addressed my all comments and suggestions.
Author Response
Thank you very much for your revision. We are very grateful to you.
Reviewer 2 Report
Thank you for improving the article as recommended.
Accept in present form.
Author Response
Thank you very much for your review and for accepting the article after improvement. We are very grateful to you.